# PSMD14 Targeting Triggers Paraptosis in Breast Cancer Cells by Inducing Proteasome Inhibition and Ca^2+^ Imbalance

**DOI:** 10.3390/ijms23052648

**Published:** 2022-02-28

**Authors:** Hong-Jae Lee, Dong-Min Lee, Min-Ji Seo, Ho-Chul Kang, Seok-Kyu Kwon, Kyeong-Sook Choi

**Affiliations:** 1Department of Biochemistry and Molecular Biology, Ajou University, Suwon 16499, Korea; katioking@ajou.ac.kr (H.-J.L.); ldonglminl@ajou.ac.kr (D.-M.L.); hisseomin22@ajou.ac.kr (M.-J.S.); 2Department of Physiology, Ajou University School of Medicine, Suwon 16499, Korea; hckang15@gmail.com; 3Brain Science Institute, Korea Institute of Science and Technology (KIST), Seoul 02792, Korea; skkwon@kist.re.kr; 4Division of Bio-Medical Science & Technology, KIST School, Korea University of Science & Technology (UST), Daejeon 34113, Korea

**Keywords:** PSMD14, paraptosis, proteasome, proteostasis, Ca^2+^ imbalance

## Abstract

PSMD14, a subunit of the 19S regulatory particles of the 26S proteasome, was recently identified as a potential prognostic marker and therapeutic target in diverse human cancers. Here, we show that the silencing and pharmacological blockade of PSMD14 in MDA-MB 435S breast cancer cells induce paraptosis, a non-apoptotic cell death mode characterized by extensive vacuolation derived from the endoplasmic reticulum (ER) and mitochondria. The PSMD14 inhibitor, capzimin (CZM), inhibits proteasome activity but differs from the 20S proteasome subunit-inhibiting bortezomib (Bz) in that it does not induce aggresome formation or Nrf1 upregulation, which underlie Bz resistance in cancer cells. In addition to proteasome inhibition, the release of Ca^2+^ from the ER into the cytosol critically contributes to CZM-induced paraptosis. Induction of paraptosis by targeting PSMD14 may provide an attractive therapeutic strategy against cancer cells resistant to proteasome inhibitors or pro-apoptotic drugs.

## 1. Introduction

The 26S proteasome consists of a 20S proteolytic core particle (CP) and one or two 19S regulatory particles (RPs). Although the proteasome inhibitors (PIs) that target 20S proteolytic CP activity, such as bortezomib (Bz) and carfilzomib, offer effective therapy for multiple myeloma (MM) patients [1,2], drug resistance often emerges, and the clinical efficacy of Bz as a single agent is limited in solid tumors [3,4]. Therefore, there is an urgent and unmet need to develop new drugs that target proteostasis through different mechanisms. PSMD14 (also known as RPN11 and POH1), a subunit of the 19S RPs, deubiquitinates substrates and prompts their degradation by the 20S proteolytic CP [5,6]. Its expression is increased and associated with poor outcomes in various cancers, including MM [7], hepatocellular carcinoma (HCC) [8], esophageal squamous cell carcinoma (ESCC) [9], lung adenocarcinoma [10], and breast cancer [11], suggesting that PSMD14 may serve as a potential prognostic marker and therapeutic target in human cancers. Recently, the PSMD14 inhibitor capzimin (CZM), a derivative of quinoline-8-thiol, stabilized a subset of proteasome substrates in treated cells and inhibited the proliferation of cancer cells [12,13]. Here, we report that the silencing and pharmacological inhibition of PSMD14 induce paraptosis, a non-apoptotic cell death mode characterized by extensive vacuolation derived from the dilations of the endoplasmic reticulum (ER) and mitochondria [14,15], in various breast cancer cells but not in the MCF-10A human breast epithelial cell line. Although we need to unravel the critical molecules involved in paraptosis, the underlying mechanisms of paraptosis reportedly include disruption of proteostasis due to proteasomal inhibition [14,15,16,17] or perturbation of sulfhydryl homeostasis [18,19,20], ion (Ca^2+^ or K^+^) imbalance [21,22], and generation of reactive oxygen species (ROS) [23,24,25]. In addition, paraptosis requires de novo protein synthesis [14,26]. This study shows that Nrf1 activation and aggresome formation, which underlie the PI resistance mechanisms of cancer cells, are not induced by PSMD14 inhibition. We found that both proteasome inhibition and intracellular Ca^2+^ imbalance critically contribute to the paraptosis induced by PSMD14 inhibition in breast cancer cells. Collectively, induction of paraptosis by targeting PSMD14 may offer a novel therapeutic strategy against cancers that have acquired resistance to PIs or pro-apoptotic drugs.

## 2. Results

### 2.1. Targeting of PSMD14 Triggers Paraptosis in MDA-MB 435S Cells

To investigate the potential of PSMD14 as a cancer target, we investigated the effects of knockdown and pharmacological inhibition of PSMD14 in MDA-MB 435S breast cancer cells. We found that siRNA-mediated suppression of PSMD14 expression progressively induced cell death accompanied by cytoplasmic vacuolation in MDA-MB 435S cells (Figure 1A,B). In addition, treatment with the PSMD14 inhibitors, quinoline-8-thiol (8-TQ) and its derivative, capzimin (CZM) [12,13], phenotypically recapitulated the effect of PSMD14 knockdown (Figure 1C,D). Furthermore, treatment with CZM dose-dependently reduced the viability and induced vacuolation prior to cell death in other breast cancer cell lines, including BT549 and MDA-MB468, but not in the MCF10A human breast epithelial cell line (Figure 1E,F). These results suggest that PSMD14 inhibition is preferentially cytotoxic to these breast cancer cells compared to normal cells. We examined the cell death mode induced by PSMD14 inhibition by employing inhibitors of various cell death modes, and we found that CZM-induced cell death or vacuolation was not affected by z-VAD (an apoptosis inhibitor) (Figure 1G,H). In addition, neither PSMD14 knockdown nor CZM did cleave caspase-3 and PARP, a substrate of caspase-3, in contrast to TRAIL, a representative inducer of apoptosis (Figure 1I). Furthermore, necrostatin1 (a necroptosis inhibitor), 3-methyladenine (an early-phase autophagy inhibitor), bafilomycin A1, or chloroquine (late-phase autophagy inhibitors) did not affect CZM-induced cell death or vacuolation (Figure 1G,H). These results suggest that apoptosis, necroptosis, and autophagy may not be critically involved in CZM-mediated cytotoxicity (Figure 1G,H). Our observation of vacuolation-associated cell death prompted us to test the possible involvement of paraptosis, a cell death mode accompanied by dilation of the ER and mitochondria [14,25,26], in the anticancer effect of PSMD14 inhibition. We found that cycloheximide (CHX), which is known to inhibit paraptosis [17,26], very effectively blocked CZM-induced cell death and vacuolation (Figure 1G,H).

In addition, confocal microscopic analysis of MDA-MB 435S sublines stably expressing fluorescence in the ER (YFP-ER) stained with Mito-Tracker Red (MTR) revealed that both CZM treatment and PSMD14 knockdown induced vacuolation originating from the ER and mitochondria prior to cell death (Figure 2A,B). Electron microscopy showed that CZM-treated cells exhibited expansion of ER-derived vacuoles and formation of megamitochondria, whereas untreated cells exhibited reticular ER and filamentous mitochondria (Figure 2C). Furthermore, CHX pretreatment effectively blocked the CZM-induced dilations of the ER and mitochondria (Figure 2D). Collectively, these results indicate that targeting of PSMD14 triggers paraptosis in MDA-MB 435S cancer cells. 

### 2.2. Proteasome Inhibition Is Not Sufficient to Induce PSMD14 Targeting-Related Paraptosis in MDA-MB 435S Cells

CZM was previously shown to stabilize proteasome substrates [13], and an increase in ubiquitylated proteins is a hallmark of proteasome inhibition [27]. We found that both CZM and the knockdown of PSMD14 increased the ubiquitylated protein levels (Figure 3A). In addition, the silencing and pharmacological inhibition of PSMD14 commonly increased the ER stress response signals, including the levels of phosphorylated eIF2α (p-eIF2α), ATF4, and CHOP. Next, we examined whether CZM-mediated proteasome inhibition is critical for this agent’s ability to induce paraptosis, and Bz, which provides an anticancer activity by inhibiting the 20S CP, can also induce paraptosis in the same cell line. Assessment of the proteasome activity employing the Ub^G76V^-GFP reporter, which contains a single uncleavable N-terminally linked ubiquitin that is attached to GFP and acts as a substrate for polyubiquitination and proteasome-mediated proteolysis [28,29], revealed a significant proteasomal inhibition by 5 μM CZM (Figure 3B), which inhibited the viability of MDA-MB 435S cells by 50% (Figure 3C). When we compared the effects of 5 μM CZM on the proteasome activity and cell viability with those of Bz, we found that the proteasome-inhibiting effect of 5 μM CZM was less potent than that of 5 nM Bz (Figure 3B), which exhibited a cytostatic effect (Figure 3C) without any morphological change (Figure 3D). Western blotting also revealed that the accumulation of ubiquitylated proteins was much lower in cells treated with 5 μM CZM than those treated with 5 nM Bz (Figure 3E). Moreover, ER stress response signals, including the levels of phosphorylated eIF2α (p-eIF2α), ATF4, and CHOP, were potently upregulated by 5 μM CZM but not by 5 nM Bz (Figure 3E). These results suggest that proteasome inhibition per se is insufficient to induce CZM-induced anticancer effect. Next, we examined whether CZM might modulate the mechanisms associated with PI resistance of cancer cells differently from Bz. The mechanisms thought to confer PI resistance to cancer cells include the upregulation of Nrf1, a transcription factor that increases the expression of proteasome subunits, and the formation of aggresome, which allows cells to escape proteotoxicity by sequestering toxic cellular aggregates [30,31]. When we compared the effect of Bz or CZM on the expressions of Nrf1 and aggresome, we found that, in contrast to Bz, CZM does not induce the upregulation of nuclear Nrf1 (Figure 3E,F). In addition, while Ub-p62 double-positive aggresomes were commonly observed in most Bz-treated cells, the scattered expression of Ub without p62 expression was detected in most CZM-treated cells, suggesting that Bz, but not CZM, induces aggresome formation (Figure 3G). These results suggest that targeting PSMD14 may have an advantage over targeting Bz, a 20S proteasome CP, in that the former may evade the mechanisms through which cancer cells acquire resistance to 20S CP-targeting PIs.

### 2.3. CZM Disrupts Intracellular Ca^2+^ Homeostasis in MDA-MB 435S Cells

Since we previously reported that perturbation of intracellular Ca^2+^ homeostasis is crucial for curcumin- or celastrol-induced paraptosis [16,22], we next investigated whether intracellular Ca^2+^ imbalance may be involved in CZM-induced paraptosis. Flow cytometric analysis using Fluo-3 (a cytosolic Ca^2+^ indicator) and Rhod-2 (a mitochondrial Ca^2+^ indicator) revealed that CZM increased both cytosolic and mitochondrial Ca^2+^ levels, which exhibited peaks at 8 h and 24 h, respectively (Figure 4A,B). Confocal microscopy showed that CZM, but not Bz, increased [2] cytosolic Ca^2+^ levels (Figure 4C) and accumulated Ca^2+^ within the dilated mitochondria (Figure 4D), and, thus, had effects similar to those of celastrol [22] (Figure 4C,D). To explore the source of the Ca^2+^ overloaded to the cytosol and mitochondria and to image subcellular Ca^2+^ dynamics, we employed G-CEPIA1er [32], cyto-RCaMP1h, and mito-RCaMP1h [33,34], which are genetically encoded fluorescent Ca^2+^ indicators that allow for the visualization of Ca^2+^ in the ER, cytosol, and mitochondria, respectively. When we treated MDA-MB 435S cells co-expressing G-CEPIA1er and Cyto-RCaMP1h with CZM to visualize ER ([Ca^2+^]_ER_) and mitochondrial matrix Ca^2+^ ([Ca^2+^]_mito_) dynamics simultaneously, we found that the dramatic decrease in ER Ca^2+^ accompanied a robust increase in cytosolic Ca^2+^ (Figure 4E and Appendix A). Imaging Ca^2+^ dynamics in cells co-expressing G-CEPIA1er and Mito-RCaMP1h, we found that mitochondrial Ca^2+^ levels were gradually and mildly increased, in contrast to the rapid and marked decrease in ER Ca^2+^. In addition, pretreatment with either BAPTA-AM, a cytosolic Ca^2+^ chelator, or Ru360, an inhibitor of the mitochondrial Ca^2+^ uniporter (MCU), effectively blocked the mitochondrial accumulation of Ca^2+^ (Figure 4F), suggesting that CZM treatment triggered the release of Ca^2+^ from ER stores into the cytosol, and that this increased Ca^2+^ is then entered into mitochondria via the MCU.

### 2.4. Both Proteasome Inhibition and Ca^2+^ Imbalance Critically Contribute to the Paraptosis Induced by PSMD14 Inhibition in MDA-MB 435S Cells

Next, we investigated the functional significance of Ca^2+^ imbalance in the paraptosis induced by VCP inhibition. We found that BAPTA-AM, but not Ru360, significantly inhibited CZM-induced cell death and vacuolation (Figure 5A,B). Interestingly, in CZM-treated cells, BAPTA-AM very effectively blocked ER dilation but not mitochondrial dilation (Figure 5C). In contrast, Ru360 had no blocking effect on these paraptotic events (Figure 5A–C). These results suggest that the release of Ca^2+^ from the ER may be critical for CZM-induced cell death, particularly in the context of ER dilation, whereas mitochondrial Ca^2+^ overload does not play a critical role in this cell death. Next, we further investigated the significance of the increase in cytosolic Ca^2+^ in CZM-induced ER stress. BAPTA-AM notably reduced CZM-induced ER stress signals, including the upregulations of p-eIF2α, ATF4, and CHOP, but not the accumulation of ubiquitylated proteins, whereas CHX pretreatment almost wholly blocked all of these stress signals (Figure 5D). These results suggest that the increase in cytosolic Ca^2+^ may critically contribute to CZM-induced ER stress and proteotoxicity, although it does not affect the proteasome activity. We also found that similar to BAPTA-AM, CHX very effectively blocked the CZM-induced increase in cytosolic Ca^2+^ levels (Figure 5E). These results suggest that CZM-induced proteasome inhibition may play an essential role in the induced Ca^2+^ imbalance, prompting us to speculate that there may be cross-modulation between the impairment of proteostasis and Ca^2+^ imbalance. Collectively, our results suggest that inhibition of PSMD14 may trigger paraptosis through the simultaneous inhibition of proteasome activity and Ca^2+^ homeostasis in MDA-MB 435S breast cancer cells.

## 3. Discussion

The targeting of proteasome 20S peptidase activity with Bz and carfilzomib has revolutionized the treatment of MM, but not all patients respond to these compounds, and those who do eventually suffer a relapse. In addition, Bz has had disappointing results when treating solid tumors [3,4]. Therefore, novel therapeutic strategies to overcome PI resistance are needed. PSMD14 is a 19S-proteasome-associated deubiquitinating enzyme that facilitates protein degradation by the 20S proteasome core particle [35]. The oncogenic roles of PSMD14 in various cancers have been associated with its effect on deubiquitinating and stabilizing various protein substrates, including ErbB2 [36], E2F1 [37,38], TGF-beta receptors, caveolin-1 [39], GRB2 [40], and SNAIL [9], suggesting that targeting PSMD14 could be a promising strategy for cancer treatment. Recently, PSMD14 inhibitors, such as o-phenanthroline (OPA) [7] and CZM [13], were reported to reduce the viability of MM cells by inducing an unfolded protein response (UPR) and stabilizing proteasome substrates, thereby exerting cytotoxicity against even Bz-resistant cancer cells [7,13]. In addition, various PSMD14 inhibitors, including OPA [7,40,41], thiolutin [41], and CZM [13,42], as well as PSMD14 knockdown [9,10,40,43,44,45,46], have demonstrated effective anticancer effects in various solid tumor cells, suggesting that targeting PSMD14 may be therapeutically relevant in a broad spectrum of cancers, including solid tumors. In the present study, we show that targeting PSMD14 may evade the mechanisms through which cancer cells acquire resistance to Bz, a PI targeting 20S proteasome CP, such as the aggresome formation and Nrf1 activation [30,31]. Therefore, these results suggest that targeting PSMD14 upstream of the 20S proteasome may be a beneficial strategy for overcoming the PI resistance of cancers and the limitations of the current 20S CP-targeting PIs in cancer therapy. However, the underlying mechanisms by which inhibition of PSMD14 exerts anticancer effects are unclear. We herein show for the first time that pharmacological or genetic inhibition of PSMD14 in several breast cancer cells induces paraptosis as a major cell death mode. Although the molecular basis of paraptosis remains to be further elucidated in detail, disruption of proteostasis (including proteasome inhibition) and Ca^2+^ imbalance have been proposed as the underlying mechanisms of paraptosis [14,15,16,17,18,19,20,21,25]. The vacuolization observed during paraptosis is believed to result from an influx of water into the ER and mitochondria, which occurs due to the increase in osmotic pressure induced by the accumulation of misfolded proteins within these organelles [14,25] or Ca^2+^ overload in mitochondria [16,21,22]. The present study shows that either pharmacological or genetic silencing of PSMD14 inhibits proteasome activity and increases cytosolic Ca^2+^ levels in MDA-MB 435S cells (Figure 3A, Figure 4A,C and Appendix A). CZM also induced mitochondrial Ca^2+^ overload (Figure 4B,D). While CHX effectively inhibited all CZM-induced paraptotic events and signals, BAPTA-AM inhibited the ability of CZM to increase Ca^2+^ in the cytosol and mitochondria and induce ER stress, ER dilation, and cell death, but did not impact the ability of CZM to induce mitochondrial dilation. These results suggest that, in addition to CZM-induced proteasome inhibition, the CZM-induced release of Ca^2+^ from the ER may critically contribute to the ER stress and ER dilation seen in breast cancer cells treated with this agent. We speculate that depletion of Ca^2+^ in the ER following CZM treatment may hamper the actions of chaperones, which depend on Ca^2+^ binding to promote folding in the ER; this may further contribute to the accumulation of misfolded proteins (ER stress), leading to the water-influx-based distention of the ER lumen [21]. The effective blocking effect of CHX on CZM-induced paraptosis may reflect the prevention or relief of the cellular proteotoxic stress derived from CZM-induced perturbations in protein degradation and Ca^2+^ homeostasis. Interestingly, RU360 did not affect CZM-induced paraptosis, despite its ability to block mitochondrial Ca^2+^ overload. These results indicate that the CZM-induced mitochondrial Ca^2+^ overload results from MCU-mediated mitochondria uptake of Ca^2+^ released from the ER Ca^2+^ but did not cause the observed mitochondrial dilation. Therefore, we cannot exclude the possibility that CZM-induced impairment of protein degradation within mitochondria may cause their dilation, as seen for the ER dilation. At present, it is not clear whether the paraptosis induced by PSMD14 inhibition is directly linked to its deubiquitinating activity. In this study, although we clearly show that PSMD14 inhibition induces Ca^2+^ imbalance, the target(s) of PSMD14 that may play a critical role in maintaining Ca^2+^ homeostasis need to be identified by further study.

Collectively, targeting PSMD14 may offer an effective therapeutic strategy to overcome the resistance of cancer cells to the 20S CP proteasome by, additionally, inducing Ca^2+^ imbalance together with proteasome inhibition. Therefore, our present results lead us to propose that inducing paraptosis by targeting PSMD14 may provide an attractive therapeutic strategy against cancer cells that exhibit resistance to the PIs and pro-apoptotic drugs.

## 4. Materials and Methods

### 4.1. Chemicals and Antibodies

Chemicals and reagents were obtained as follows: CZM from, 8-TQ from, Bz from Selleckchem (Houston, TX, USA); TRAIL from KOMABIBIOTECH (Seoul, South Korea); Ru360 from Calbiochem (Millipore Corp., Billerica, MA, USA); MitoTracker-Red (MTR), propidium iodide (PI), Fluo-3-AM, Rhod-2-AM, and 4′,6-diamidino–2-phenylindole (DAPI) from Molecular Probes (Eugene, OR, USA); z-VAD-fmk from R&D Systems (Minneapolis, MN, USA); necrostatin-1, 3-methyladenine (3-MA), bafilomycin A1, 2-bis(o-amino phenoxy)ethane-N, N, N’N′-tetraacetic acid acetoxymethyl ester (BAPTA-AM), celastrol, and cycloheximide (CHX) from Sigma-Aldrich (St. Louis, MO, USA). The following primary antibodies were used: Ub from Santa Cruz (Dallas, TX, USA), p62 from BD Biosciences (San Jose, CA, USA); p-eIF2α, eIF2α, ATF4, Nrf1, and CHOP/GADD153 from Cell Signaling Technology (Danvers, MA, USA); caspase-3 from Enzo Life Science (Farmingdale, NY, USA); PARP from Abcam (Cambridge, UK). The secondary antibodies, including rabbit IgG HRP, mouse IgG HRP, rabbit Alexa Fluor 488, and mouse Alexa Fluor 594, were obtained from Molecular Probes.

### 4.2. Cell Culture

MDA-MB 435S, MDA-MB 231, MDA-MB 468, and MCF-10A cells were purchased from American Type Culture Collection (ATCC, Manassas, VA, USA). MDA-MB 435S cells were cultured in low-glucose DMEM, MDA-MB 231, and MDA-MB 468 cells in RPMI-1640 medium supplemented with 10% fetal bovine serum and 1% antibiotics (GIBCO-BRL, Grand Island, NY, USA). MCF-10A cells were maintained in DMEM/F12 medium supplemented with 5% horse serum, insulin, hydrocortisone, and cholera toxin (Calbiochem, Millipore Corp., Billerica, MA, USA). Cells were incubated in 5% CO_2_ at 37 °C

### 4.3. Cell Viability Assay

Cells were cultured in 48-well plates and treated as indicated. The cells were then fixed with methanol/acetone (1:1) at −20 °C for 5 min, washed three times with PBS, and stained with propidium iodide (PI; final concentration, 1 μg/mL) at room temperature for 10 min. The plates were imaged on an IncuCyte device (Essen Bioscience, Ann Arbor, MI, USA) and analyzed using the IncuCyte ZOOM 2016B software. The processing definition of the IncuCyte program was set to recognize attached (live) cells by their red-stained nuclei. The percentage of live cells was normalized to that of untreated control cells (100%).

### 4.4. Morphological Examination of the ER and Mitochondria

YFP-ER cells were established as described previously [17]. After treatments, YFP-ER cells were stained with 100 nM MTR for 10 min, and morphological changes of the ER and mitochondria were observed under a K1-Fluo confocal laser scanning microscope (Nanoscope Systems, Daejeon, Korea).

### 4.5. Immunoblot Analysis

Cells were washed in PBS and lysed in boiling sodium dodecyl sulfate–polyacrylamide gel electrophoresis (SDS–PAGE) sample buffer (62.5 mM Tris (pH 6.8), 1% SDS, 10% glycerol, and 5% β- mercaptoethanol). The lysates were boiled for 5 min, separated by SDS–PAGE, and transferred to an Immobilon membrane (Millipore, Billerica, MA, USA). After nonspecific binding sites were blocked for 1 h using 5% skim milk, the membranes were incubated for 2 h with specific antibodies. Membranes were then washed three times with Tris-Buffered Saline Tween-20 (TBST) and incubated further for 1 h with horseradish peroxidase-conjugated anti-rabbit, -mouse, or -goat antibody. Visualization of protein bands was accomplished using ECL (Amersham Life Science, Amersham, UK). β-actin was used as a loading control.

### 4.6. Immunofluorescence Microscopy

After treatments, the cells were fixed with 4% paraformaldehyde for 10 min at RT and blocked in 5% BSA in PBS for 30 min. Fixed cells were incubated overnight at 4 °C with primary antibodies diluted in PBS, washed three times in PBS, and incubated for 1 h at room temperature with anti-mouse or anti-rabbit Alexa Fluor 488, 594, and 647 (1:1000, Molecular Probes). Slides were mounted with ProLong Gold antifade mounting reagent (Molecular Probes), and cell staining was visualized with the K1-Fluo confocal laser scanning microscope.

### 4.7. Imaging of Subcellular Localization of Ca^2+^

To measure cytosolic Ca^2+^ levels ([Ca^2+^]_cyto_), treated cells were incubated with 100 nM Fluo-3-AM at 37 °C for 20 min, washed with HBSS (without Ca^2+^ or Mg^2+^), and analyzed immediately under a K1-Fluo confocal laser-scanning microscope. To measure mitochondrial Ca^2+^ levels ([Ca^2+^]_mito_), treated cells were incubated with 100 nM Rhod-2-AM at 37 °C for 30 min, washed with HBSS (without Ca^2+^ or Mg^2+^), and then analyzed under a K1-Fluo confocal laser-scanning microscope. To simultaneously observe the CZM-induced changes in Ca^2+^ levels in the ER and cytosol, we co-transfected MDA-MB 435S cells with the plasmids pCAG G-CEPIA1er and pCAG cyto-RCaMP1h. To monitor CZM-induced changes in Ca^2+^ levels in the ER and mitochondria simultaneously, we cotransfected MDA-MB 435S cells with the plasmids pCAG G-CEPIA1er and pCAG mito-RCaMP1h. Transfected cells were cultured in 12-well plates and treated with 5 μM CZM for the indicated time points, and all the images were captured with a K1-Fluo confocal laser scanning microscope (Nanoscope Systems, Daejeon, Korea) at the following excitation/emission wavelengths: 488 nm/525 ± 50 nm for G-CEPIA1er and Fluo-3-AM, and 561 nm/593 ± 46 nm for cyto-RCaMP1h, mito-RCaMP1h, and Rhod-2-AM. Images were analyzed using Image J/Fiji software. Fluorescence intensities were corrected for background fluorescence (F0) by measuring the non-fluorescent area. F0 values were defined by averaging 10 frames before stimulation and used for normalization, and F of ROI was obtained by averaging intensities of pixels inside the ROI. ΔF/F = (F − F0)/F0 was calculated in each pixel at each time.

### 4.8. Small Interfering RNA-Mediated Knockdown of PSMD14

The specific siRNA targeting PSMD14 (siPSMD14; J-006024-05, 06, 07, 08) and the Negative Universal Control siRNA (siNC) were purchased from Thermo Dharmacon and Invitrogen, respectively. Transfection was performed using the RNAiMAX reagent (Invitrogen, Carlsbad, CA, USA) in accordance with the instructions of the manufacturer.

### 4.9. Transmission Electron Microscopy

Cells were prefixed in Karnovsky’s solution (1% paraformaldehyde, 2% glutaraldehyde, 2 mm calcium chloride, 0.1 m cacodylate buffer, pH 7.4) for 2 h and washed with cacodylate buffer. Post-fixing was carried out in 1% osmium tetroxide and 1.5% potassium ferrocyanide for 1 h. After dehydration with 50–100% alcohol, the cells were embedded in Poly/Bed 812 resin (Pelco, Redding, CA, USA), polymerized, and observed under the electron microscope (Sigma 500, Zeiss, Oberkochen, Germany).

### 4.10. Measurement of Proteasome Activity Employing Ub^G76V^-GFP

Cells transfected with Ub^G76V^-GFP were cultured for 48 h in a 24-well plate and treated with the indicated agents for 8 h. The plates were imaged on an IncuCyte device (Essen Bioscience, Ann Arbor, MI, USA), and the fluorescence intensities were analyzed using the IncuCyte ZOOM 2016B software. 

### 4.11. Statistical Analysis

All the data are presented as the mean ± SD (standard deviation). GraphPad Prism software (GraphPad Software Inc, San Diego, CA, USA) was used to perform the statistical analyses. The normality of data was assessed by the Kolmogorov–Smirnov test or Pearson test, and equal variance was assessed using Bartlett’s test. For a normal distribution, statistical differences were determined using analysis of variance (ANOVA) followed by Bonferroni’s multiple comparison test.

## Figures and Tables

**Figure 1 ijms-23-02648-f001:**
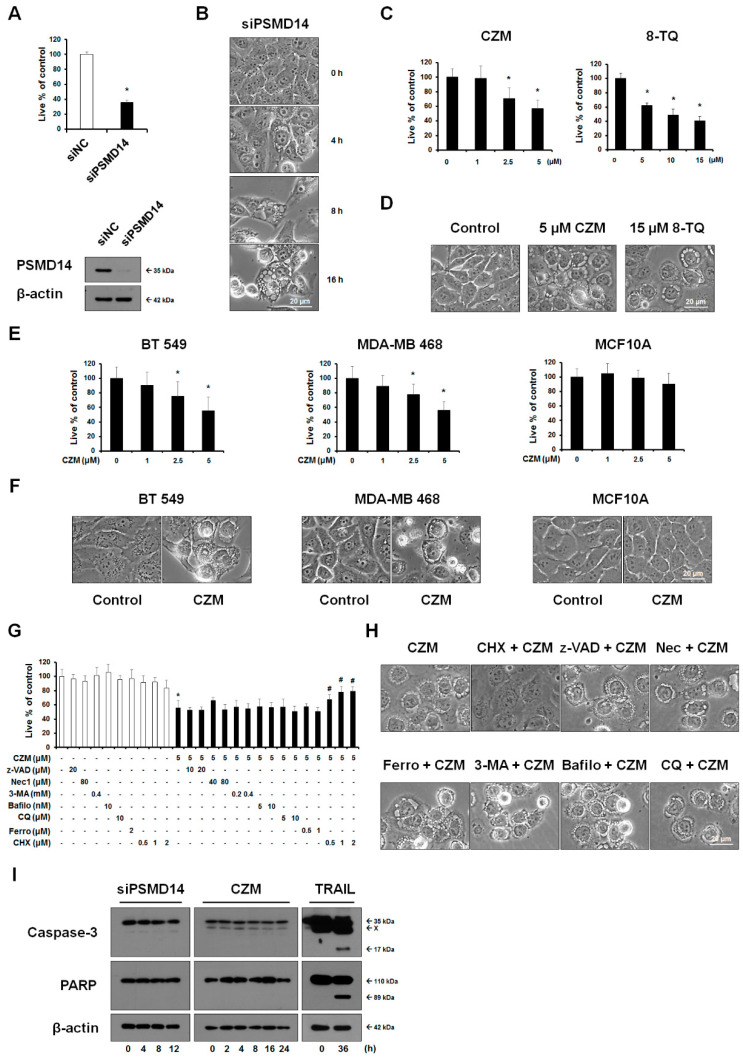
Targeting PSMD14 induces cell death accompanied by vacuolation in breast cancer cells but not in normal breast cells. (**A**,**B**) MDA-MB 435S cells were transfected with the siRNA against PSMD14. (**A**) Cellular viability was assessed, as described in Materials and Methods, and knockdown of PSMD14 was confirmed by Western blotting. * *p* < 0.001 vs. siNC-transfected cells. (**B**) Cells were observed by phase-contrast microscopy. Bars, 20 μm. (**C**,**D**) Cells were treated with the indicated concentrations of CZM for 12 h or 8-TQ for 24 h. (**C**) Cellular viability was assessed using IncuCyte, as described in Materials and Methods. * *p* < 0.001 vs. untreated cells. (**D**) Cells were observed by phase-contrast microscopy. Bars, 20 μm. (**E**) Cells were treated with the indicated concentrations of CZM for 24 h, and cellular viability was assessed using IncuCyte. * *p* < 0.001 vs. control. (**F**) Cells were treated with 5 μM CZM for 12 h and observed by phase-contrast microscopy. Bars, 20 μm. (**G**,**H**) MDA-MB 435S cells pretreated with the indicated inhibitors of various cell death modes were further treated with 5 μM CZM for 24 h (**G**) or 12 h (**H**). (**G**) Cellular viability was assessed as described above. * *p* < 0.001 vs. control, # *p* < 0.001 vs. CZM-treated cells. (**H**) Cells were observed by phase-contrast microscopy. Bars, 20 μm. (**I**) MDA-MB 435S cells were transfected with the siRNA against PSMD14 or treated with 5 μM CZM or 200 ng/mL TRAIL for the indicated time points. Western blotting of caspase-3 and PARP was performed using β-actin as a loading control.

**Figure 2 ijms-23-02648-f002:**
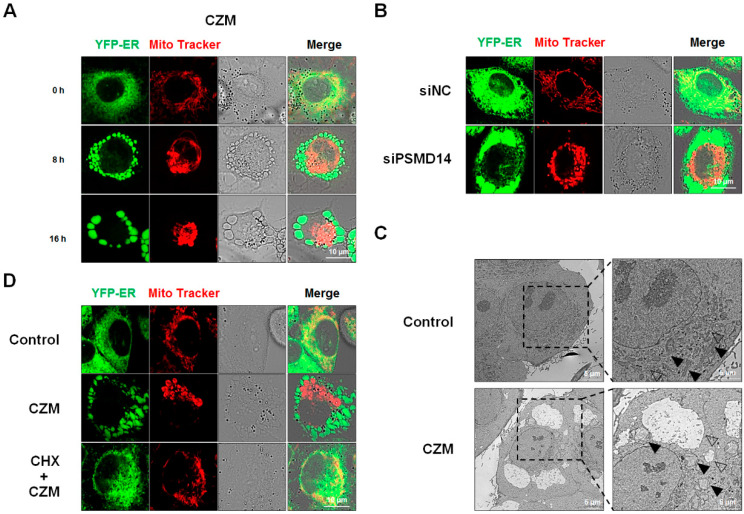
Targeting PSMD14 induces paraptosis in MDA-MB 435S cells. (**A**,**B**) YFP-ER cells were treated with 5 μM CZM for the indicated time durations (**A**) or transfected with siRNA against PSMD14 for 36 h (**B**) and stained with MitoTracker-Red. Cells were observed under the confocal microscope. Bars, 10 μm. (**C**) MDA-MB 435S cells treated with 5 μM CZM for 12 h were subjected to electron microscopy. Bars, 5 μm. (**D**) YFP-ER cells pretreated with or without 1 μM CHX were further treated with 5 μM CZM for 12 h, stained with MitoTracker-Red, and observed by confocal microscopy. Bars, 10 μm.

**Figure 3 ijms-23-02648-f003:**
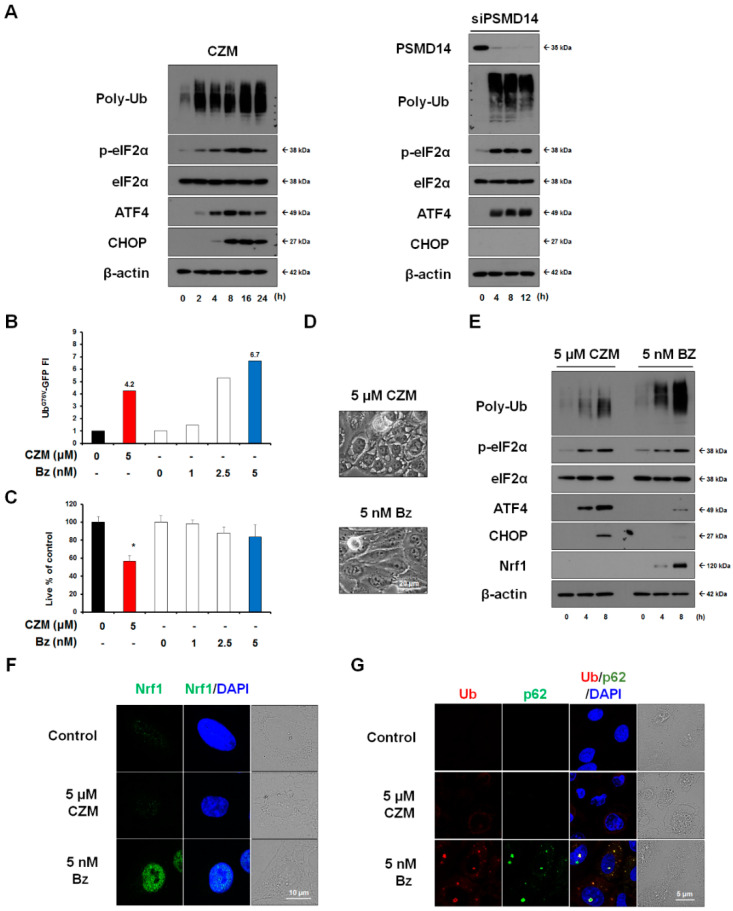
Proteasome inhibition alone is not sufficient to explain the anticancer effect of PSMD14. (**A**) Cell extracts were prepared from MDA-MB 435S cells treated with 5 μM CZM or transfected with the siRNA against PSMD14 for the indicated time durations. Western blotting of the indicated proteins was performed using β-actin as a loading control. (**B**) MDA-MB 435S cells transfected with Ub^G76V^-GFP were treated with the indicated concentrations of CZM or Bz for 12 h, and the fluorescence intensity was assessed as described in Materials and Methods. (**C**) Viability was assessed using IncuCyte in MDA-MB 435S cells treated with CZM or Bz at the indicated concentrations for 24 h. * *p* < 0.001 vs. untreated cells. (**D**) MDA-MB 435S cells were treated with 5 μM CZM or 5 nM Bz for 12 h and observed by phase-contrast microscopy. Bars, 20 μm (**E**). MDA-MB 435S cells were treated with 5 μM CZM or 5 nM Bz for the indicated time durations, and Western blotting of the indicated proteins was performed using β-actin as a loading control. (**F**,**G**) MDA-MB 435S cells were treated with 5 μM CZM or 5 nM Bz for 12 h, and immunocytochemistry of the indicated proteins was performed. Bars, 10 μm (**F**) or 5 μm (**G**).

**Figure 4 ijms-23-02648-f004:**
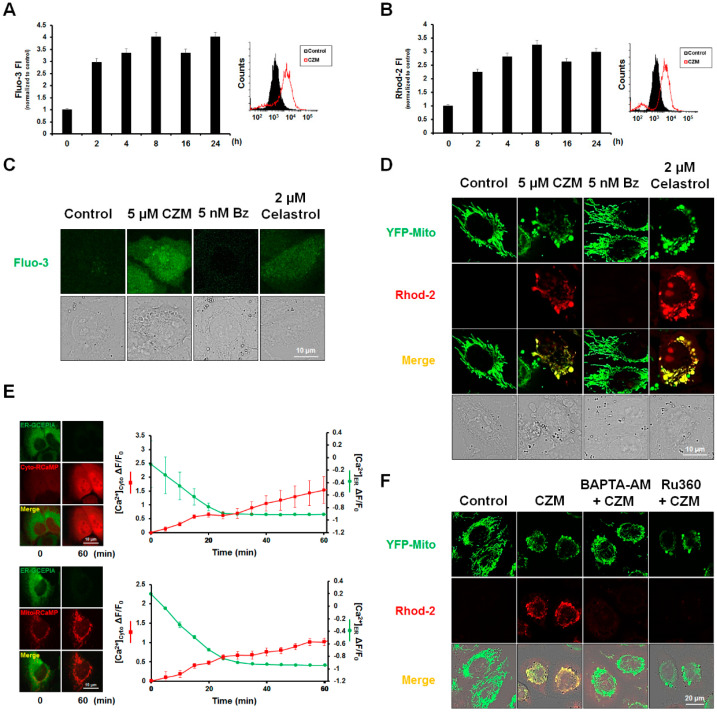
PSMD14 inhibition induces intracellular Ca^2+^ imbalance in MDA-MB 435S cells. (**A**,**B**) MDA-MB 435S cells were treated with 5 μM CZM for the indicated time durations, stained with Fluo-3 (**A**) or Rhod-2 (**B**) and processed for FACS analysis. The results of FACS analysis after treatment with 5 μM CZM for 8 h are depicted in the graphs. (**C**) MDA-MB 435S cells treated with the indicated chemicals were observed under the confocal microscope. Bars, 10 μm. (**D**) YFP-Mito cells treated with the indicated chemicals were stained with Rhod-2 and observed under the confocal microscope. Celastrol was used as a positive control to confirm the increase in Ca^2+^ in the cytosol and mitochondria. Bars, 10 μm. (**E**) MDA-MB 435S cells transfected with the plasmids encoding G-CEPIA1er and cyto-RCaMP1h or cells transfected with those G-CEPIA1er and mito-RCaMP1h were treated with 5 μM CZM for the indicated time durations. Subcellular Ca^2+^ dynamics were monitored under the confocal microscope, and representative images were displayed. Bars, 10 μm. Time course Ca^2+^ signal in the ER, cytosol, and mitochondria in CZM-treated cells was analyzed using Image J/Fiji software. (**F**) YFP-ER cells pretreated with 10 μM BAPTA-AM or 20 μM Ru360 were further treated with 5 μM CZM for 8 h and stained with Rhod-2. Cells were observed under the confocal microscope. Bars, 20 μm.

**Figure 5 ijms-23-02648-f005:**
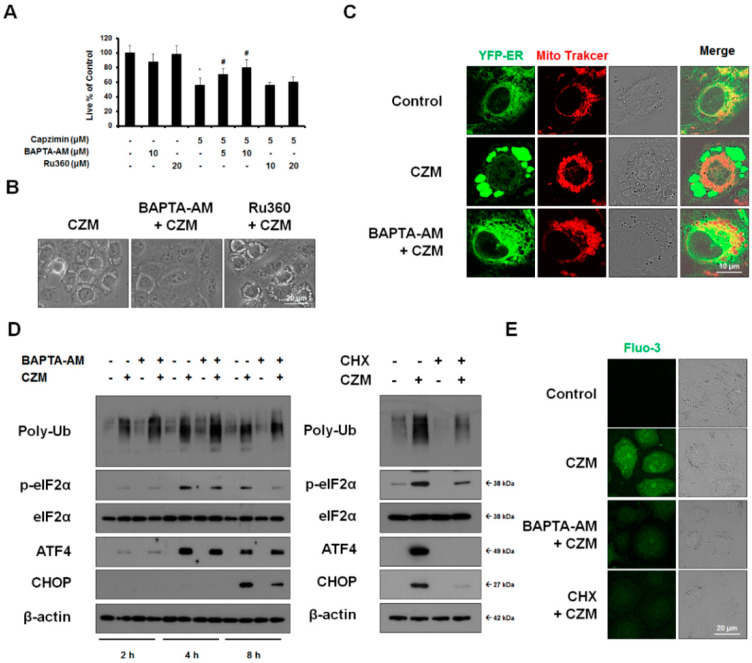
Disruption of both proteostasis and Ca^2+^ homeostasis is required for the paraptosis induced by PSMD14 inhibition in MDA-MB 435S cells. (**A**,**B**) MDA-MB 435S cells pretreated with 10 μM BAPTA-AM or 20 μM Ru360 were further treated with 5 μM CZM for 24 h or 12 h. (**A**) Cellular viability was assessed using IncuCyte, as described above * *p* < 0.001 vs. untreated cells, # *p* < 0.001 vs. CZM-treated cells. (**B**) Cells were observed by phase-contrast microscopy. Bars, 20 μm. (**C**) YFP-ER cells were pretreated with 10 μM BAPTA-AM or 20 μM Ru360 were further treated with 5 μM CZM for 8 h and stained with MitoTracker-Red. Cells were observed under the confocal microscope. Bars, 10 μm. (**D**,**E**) Cells pretreated with 5 μM BAPTA-AM or 1 μM CHX were further treated with 5 μM CZM for the indicated time durations or 8 h. (**D**) Western blotting of the indicated proteins was performed using β-actin as a loading control. (**E**) Cells were stained with Fluo-3 and observed by confocal microscopy. Bars, 20 μm.

## Data Availability

The data presented in this study are available on request from the corresponding author.

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
