# Peer review of "PSMD14 Targeting Triggers Paraptosis in Breast Cancer Cells by Inducing Proteasome Inhibition and Ca2+ Imbalance"

_ijms, 2022, doi:10.3390/ijms23052648_

Round 1

Reviewer 1 Report

In the current manuscript, the authors have shown that targeting of PSMD14 can induce paraptosis in breast cancer cells through proteasome inhibition and calcium imbalance. They provide sufficient and clear data to demonstrate ER and mitochondria vacuolation known to be associated with paraptosis as well as the blockage by CHX. They further show that proteasome inhibition, without the upregulation of Nrf1, and intracellular calcium imbalance are involved in the mechanisms of PSMD14 inhibition-induced paraptosis. Overall, the data are clear and provide sufficient support for the hypothesis of the manuscript.

For very minor edits, in lane 115 Figure 2A should be corrected to 3A and in lane 242 SMD14 should be corrected to PSMD14.

Although I consider the manuscript appropriate to be accepted in the present form, if the authors would like to more strongly emphasize the potential role of PSMD14 targeting as a method to overcome Bz resistance, I suggest that they may try the combinatorial treatment of CZM and Bz on cells or CZM treatment to Bz-resistance cells.

Overall, I believe the manuscript provides clear and sufficient data to support the overall hypothesis and that is can be accepted in the present form.

Reviewer 2 Report

In the study, the authors found that PSMD14 inhibitors induced a specialized cell death paraptosis with vacuolar degeneration via an increase in intracellular Ca2+. The results are potentially interesting, but the only drawback is that the actual mechanism of cell death is not fully understood. I have some concerns that should be addressed by the authors. Specific comments are as follows.

Major points:

  1. The authors confirmed that CZM but not BZ caused ER stress, but it is well known that apoptosis occurs under ER stress. Similarly, BZ is well known to cause apoptosis in various cancer cell lines. The authors should clarify how the execution of cell death is different by examining the differences in downstream apoptotic signals, such as activation of caspase-3.
  2. The authors claim that CZM did not induce aggresome formation, but they do not provide data to support this. Since CZM markedly increased the amount of intracellular polyubiquitinated proteins (Figure 3), the authors should stain intracellular Ub to verify the presence of aggresome formation.

Minor points

  1. The title of the manuscript includes “Cancer cells”, but most of the data is from breast cancer cell line. It should be “breast cancer”.
  2. In western blotting data, the authors should indicate the position of a molecular weight marker above and/or below the band(s) of interest.
  3. English should be carefully revised by a native English speaker or a professional English editing service.

Round 2

Reviewer 2 Report

The authors have addressed most of my concerns and I have no further points.